# Lower Limb Muscles’ Activation during Ascending and Descending a Single Step-Up Movement: Comparison between In water and On land Exercise at Different Step Cadences in Young Injury-Free Adults

**DOI:** 10.3390/healthcare11030441

**Published:** 2023-02-03

**Authors:** Billy C. L. So, Manny M. Y. Kwok, Nakita W. L. Lee, Andy W. C. Lam, Anson L. M. Lau, Allen S. L. Lam, Phoebe W. Y. Chan, Shamay S. M. Ng

**Affiliations:** Gait and Motion Analysis Laboratory, Department of Rehabilitation Sciences, The Hong Kong Polytechnic University, Hong Kong, China

**Keywords:** muscles activation, stepping exercises, water immersion

## Abstract

(1) Background: Forward step-up (FSU) simulates the stance phase in stair ascension. With the benefits of physical properties of water, aquatic FSU exercise may be more suitable for patients with lower limb weakness or pain. The purpose of this study is to investigate the effect of progressive steps per min on the surface electromyography (sEMG) of gluteus maximus (GM), biceps femoris (BF), rectus femoris (RF), and gastrocnemius (GA), when performing FSU exercise with different steps per min in water and on land. (2) Methods: Participants (N = 20) were instructed to perform FSU exercises at different steps per min (35, 60, and 95 bpm) in water and on land. The sEMG of the tested muscles were collected. The percentage maximum voluntary isometric contraction (%MVIC) of GM, RF, GA and BF at different environments and steps per min was compared. (3) Result: There was a statistically significant difference of %MVIC of RF at all steps per min comparisons regardless of the movement phases and environments (*p* < 0.01, except for descending phases of 35 bpm vs. 60 bpm). All tested muscles showed a statistically significant lower muscle activation in water (*p* < 0.05) (4) Conclusion: This study found that the %MVIC of the tested muscle in both investigated environments increase as steps per minute increases. It is also found that the movement pattern of FSU exercise activates RF the most among all the tested muscles. Muscle activation of all tested muscles is also found to be smaller in water due to buoyancy property of water. Aquatic FSU exercise might be applicable to patients with lower limb weakness or knee osteoarthritis to improve their lower limb strength.

## 1. Introduction

Aquatic exercises are gaining popularity in musculoskeletal rehabilitation, such as knee osteoarthritis, and are recommended to patients who are unable to exercise on land due to physical limitations, such as pain and swelling [1]. The physical properties of water, such as buoyancy, hydrostatic pressure, and drag force, may account for the recommendations made by clinicians [2]. Buoyancy is proposed to be able to reduce compressive force on joint since it can reduce body weight with different immersion depth, for example, immersed to xiphisternum can offload 50–76% of body weight [3,4]. Therefore, buoyancy may reduce joint pain when exercising in water. Hydrostatic pressure, which is directly proportional to immersion depth, could act as a force that aids the resolution of swelling in body parts [5]. Drag force, which is affected by the viscosity and the speed of movement, can be utilized as resistance when a body segment moves relative to water [2]. Drag force is commonly progressed by altering the speed or surface area of limbs [6].

During aquatic exercise, higher speed of moving body parts significantly increases the training load, such that doubling the speed will increase drag force by four times [7]. Therefore, clinicians usually manipulate the parameter of speed, or steps per min to alter the resistance exerted on the body parts to optimize rehabilitation outcomes [1]. Compared to land-based exercise, speed and drag force are unique factors in precise aquatic exercise prescription since they both change the load and movement tasks. The velocity of particular muscle contractions is a crucial factor in specific muscle trainings and performance adaptations which warrant greater attention in aquatic rehabilitation. Previous studies] have investigated the effect of speed on different muscle groups in water and land [8,9,10,11]. Speed can be viewed as one of the progression components in aquatic resistance exercise. However, relatively limited studies have investigated the effect of speed on lower limb strengthening exercise, especially unilateral weight-bearing exercise.

Unilateral weight-bearing exercise is commonly prescribed by clinicians in order to stimulate functional muscle recruitment patterns required for daily living and sports, with forward step-up (FSU) exercise being the most common form [12,13]. FSU exercise can be a progression of bilateral weight-bearing exercise as it requires higher muscle activation to achieve [14]. Moreover, FSU exercise is an functional exercise mimicking stair-climbing, which requires simultaneous coordination of hip, knee, and ankle musculature [15]. Aquatic FSU exercise may aid individuals with lower limbs pain or weakness to perform more challenging functional activities with less difficulty.

The potential benefits of FSU exercise are not yet been confirmed and the associated perceptive responses (i.e., the influences of speed and environment) are yet to be known. Therefore, the aim of this study was to investigate the effect of progressive steps per min on the muscle activation of gluteus maximus (GM), rectus femoris (RF), biceps femoris (BF), and gastrocnemius (GA) performed in water and on land. According to Zimmermann et al. [16], three steps per min were selected: 35, 60, and 95 beats per minute (bpm). We hypothesized that muscle activation of the target muscles was lower in water than on land, and they increased as the steps per min increased. The result of this study will aid physiotherapists to customize parameters of FSU exercise to patients in musculoskeletal rehabilitation.

## 2. Materials and Methods

### 2.1. Study Design

This study was a cross-sectional study to compare surface electromyography (sEMG) activity of performing FSU exercise in water or on land at three different steps per min. As shown in Figure 1, 26 participants were assessed for eligibility. The included participants (n = 20) were randomly allocated to Group A and Group B in which each group consisted of five males and five females. To eliminate directionality problem, group A participants were asked to perform land trial followed by water trial while Group B participants were asked to perform water trial followed by land trial. The two trials were completed on separate days with sufficient rest in between.

### 2.2. Sample Size Planning

The sample size was calculated based on the primary outcome of a previous study [17] comparing sEMG of lower extremities between land and water step-up exercise. Using the G* Power software version 3.0.10 and based on the effect size of f = 0.79 between the exercise groups obtained, the primary outcome sEMG of lower extremities assumed a 5% type I error and 80% power. The sample size computed was 15 or more participants. Considering an estimated 30% attrition rate, the total enrolled sample size required to ensure adequate statistical power was 20.

### 2.3. Participants

Participants were recruited by convenient sampling in Hong Kong Polytechnic University. Twenty healthy young adults (10 females and 10 males) aged 18 to 30 participated in this study. Individuals were excluded from this study if they had (1) any musculoskeletal, bone, joint, cardiac, and pulmonary diseases, any infectious diseases, skin conditions, and any known hip or knee injuries (included previous hip or knee surgeries) in recent two years, and (2) any contraindications to aquatic exercises, or prior exposure to aquatic-based exercises. Prior to participation, participants’ demographic information including resting heart rate, blood pressure, age, height, weight, BMI, and leg dominance (i.e., the foot used to kick a ball) were obtained. Participants were informed of the nature of the study and signed a consent form prior to voluntary participation. This study was approved by the Departmental Research Committee of the Hong Kong Polytechnic University’s Department of Rehabilitation Sciences (Reference Number: HSEARS 20220204005).

### 2.4. Experimental Set-Up

The trials were videotaped using waterproof camera GoProHERO3 at 90 frames/s. The camera was placed 1.5 m away from the participants and positioned at patellofemoral joint level to prevent any angulation of the video. sEMG activities were recorded using a 16-channel sEMG system (Infinity Mini Wave waterproof, Cometa, Milan, Italy), and a customized data logger at 1000 Hz sampling rate. The sEMG signals were then exported using EMGandMotionTools version 8.3.4.0 (Cometa, Milan, Italy).

### 2.5. Procedures

The standardized procedures of the study were explained to the participants, and they were as follows:

#### 2.5.1. Skin Preparation, Electrode Placement and Joint Markers

The muscle activation of our target muscles was evaluated by sEMG. To minimize the impedance, the required skin areas for electrode placement were shaved, handled with abrasive material (3M Red Dot Trace Prep), and cleaned with alcohol swab (70% isopropyl). According to previous studies [16,18], the electrodes were applied to the skin of the dominant leg of participants as follows: GM’s electrode: at a point half of the distance between the greater trochanter of the femur and the superior end of the gluteal cleft (Figure 2a); RF’s electrode: midway between the anterior superior iliac spine and the superior edge of patella (Figure 2b); BF’s electrode: midway between ischial tuberosity and medial joint line of the knee (Figure 2c); GA’s electrode: at a point one-fourth of the distance from the medial knee joint line to the base of the calcaneus (Figure 2d). In order to record sEMG signals under water, waterproof technique was adopted by using tegaderm (3M™ Tegaderm™ Transparent Film Roll 16002). Three bony landmarks with markers of 3 cm in diameter were attached over greater trochanter of femur, lateral epicondyle of femur, and lateral malleolus for kinematic tracking (Figure 2e).

#### 2.5.2. Pre-Trial: MVIC Tests

Participants performed MVIC tests for each muscle group on land to normalize sEMG data recorded during FSU exercise on land and in water. A 2 min rest was given between each MVIC. Three 5-s MVICs were recorded for each muscle group tested. The order of MVIC tests was GM, RF, BF, and, finally, GA. According to Zimmermann et al. [16] and Yuen et al. [18], MVIC tests were performed as follows: GM MVIC was obtained when the participants extended their hips maximally and maintained 90° of knee flexion in prone; RF MVIC was obtained with the participants seated on a plinth and extending the knee at a secured angle of 45–50° of knee flexion; BF MVIC was obtained when the participants stood on the non-dominant leg and performed isometric knee flexion with the dominant knee flexed at 90°. Participants were allowed to support themselves against the wall using their arms for balance. GA MVIC was obtained with the participants seated with their hips flexed and feet in front of them, then plantar flexed maximally with their knees flexed 20° and feet resting on a stable stool. Consistent verbal encouragement was provided during all MVIC.

#### 2.5.3. FSU Exercise Standard Protocol

Standardized instructions of performing the required movement were given to participants. The participants were instructed to perform FSU exercise with a plyometric box of 21 cm high without using their hands for balance. Each participant performed a 5 min warm-up stepping exercise. After warm-up, 1 min familiarization session was conducted to ensure that the participants accommodated to the test conditions. Participants received researchers’ guidance throughout the familiarization session. In the test session, the order of test steps per min (35, 60, and 95 bpm) was randomly assigned to each participant. Three sets of eight repetitions for each step per min (35, 60, and 95 bpm) were performed according to the assigned order. Two min rest was given between each set and 5 min rest was given between each steps per min. Additional verbal cues were given to participants in order to maintain synchronization with steps per min and accuracy of the standard movement. The instructions of the FSU exercise are as follows: The participants started with the foot of the dominant leg placed entirely on the plyometric box with their weight shifted to dominant leg only (Figure 3a). For the ascending phase, they extended the dominant hip and knee to move the body to a standing position (Figure 3b). They were instructed not to use the non-dominant leg to push off. For the descending phase, they descended with the dominant leg and returned to starting position (Figure 3c). They were asked to watch a video clip combining visual and auditory cues while performing FSU exercise.

Regarding the temperature, land trial was maintained at room temperature (25 °C) while water trial was maintained at the indoor swimming pool temperature (28 °C). The water level was set at chest level in starting position. Participants were required to stand on either a 15 cm or a 25 cm tall platform if the water level was above the chest level.

### 2.6. Outcome Measurements

Normalized muscle activation (%MVIC) was obtained through dividing the recorded muscle activation of the tested muscles individually by the maximum muscle activation values estimated from MVIC tests.

### 2.7. Data Processing

Raw sEMG signals were processed by bandpass filter (at 20 Hz to 300 Hz) and root-mean-square sliding window (50 ms time constant) (MatLab R2020a; Mathematical computing software, Natick, MA, USA). With reference to Mercer et al. [19], 6 to 12 steps from the 24 steps from each steps per min were selected for analysis. The middle four of the eight steps in each set were selected. For the kinematic data, a customized program was used to determine the period of the middle four FSU movements. Knee angles at the corresponding time were analyzed from markers on participants in the videos taken using motion-tracking software Kinovea (v.0.9.5) (Kinovea, Bordeaux, Nouvelle Aquitaine, France). The initial time for the third step and final time for sixth step will be marked to synchronized with sEMG data for statistical analysis. Amplitudes of EMG signal for the four targeted muscles were calculated and averaged. Raw MVIC data were filtered and smoothed in the same way as raw sEMG signals. MVIC values of the three bursts of contractions were first calculated into three separate means. The greatest mean MVIC value among the three bursts was selected as the MVIC value of the tested muscles. Mean sEMG amplitudes for the ascending and descending phases of the four FSU movements were normalized to these MVIC values and expressed as %MVIC.

### 2.8. Statistical Analysis

To examine the difference in sEMG activity between aquatic and land environments, cadences and between ascending and descending phases of FSU exercise. The ANOVA three-way was performed for each muscle, using the three main factors (environment, cadence, and phase) and their interactions. The statistical assumptions of normality and sphericity for using the repeated measures ANOVA were tested. In the first place, a descriptive analysis of the main anthropometric variables of the participants and of the maximum activation registered in each of the muscles analyzed in the present study (mean, standard deviation and difference) was conducted. Additionally, each variable was compared (muscle activity of GM, RF, BF, and GA (% MVIC)) between the two environments, at different cadences and various phases and their interactions. For all statistical comparisons, *p* level was set to ≤0.05. Subsequently, an analysis was made regarding the degree of contribution of each of the muscles observed during various cadences and phases. The Bonferroni test was used when there was a statistically significant difference. The effect size was calculated via the partial eta squared with 0.01 indicated a small effect, 0.06 indicated a medium effect, and 0.14 a large effect [20]. All statistical analyses were performed using IBM SPSS Statistics for Windows, Version 26.0 (IBM Corp., Armonk, NY, USA).

## 3. Results

All participants completed the sessions. There were no adverse effects or safety concerns raised in interventions. Table 1 shows the descriptive characteristics of participants.

### 3.1. Comparison of Steps per min during FSU Exercise

Figure 4 shows the change of mean % MVIC of all target muscles. The result indicated a significant difference between the mean %MVIC of all muscles, except for GA, at 35 bpm, 60 bpm, and 95 bpm in both phases on land and in water, respectively.

For the comparison between 35 and 60 steps per minute on land, the mean %MVIC of RF at 35 steps per minute was significantly lower than that of 60 steps per minute (*p* < 0.01). Regardless of movement phases on land, the mean %MVIC of GM and RF shows no statistically significant difference, except for the descending phase of GM.

For the comparison between 35 and 95 steps per minute on land, the mean %MVIC of GM, RF and BF at 35 steps per minute was significantly lower than that of 95 steps per minute (*p* < 0.01). For the RF, it shows a maximal increase of 44.4% MVIC in the ascending phase.

For the comparison between 60 and 95 steps per minute on land, the mean %MVIC of RF and BF at 60 steps per minute was significantly lower than that of 95 steps per minute (*p* < 0.01). For GM, regardless of movement phases on land, the mean %MVIC of GM shows no statistically significant difference.

For the comparison between 35 and 60 steps per minute in water, mean %MVIC of RF at 35 steps per minute was significantly lower than that of 60 steps per minute (*p* < 0.01). Regardless of movement phases in water, the mean %MVIC of GM and RF shows no statistically significant difference, except for the descending phase of GM.

For the comparison between 35 and 95 steps per minute in water, the mean %MVIC of GM, RF, and BF at 35 steps per minute was significantly lower than that of 95 steps per minute (*p* < 0.01). For RF shows a maximal increase of 68.9% MVIC in the ascending phase at the comparison of 35 and 95 bpm in water.

For the comparison between 60 and 95 steps per minute in water, the mean %MVIC of RF at 60 steps per minute was significantly lower than that of 95 steps per minute (*p* < 0.01). For GM and BF, regardless of movement phases on land, the mean %MVIC of these muscles shows no statistically significant difference.

### 3.2. Comparison of Environments during FSU Exercise

Table 2 compares the tested muscle activation in different environments. In general, all tested muscles showed lower muscle activation in water when compared to land (*p* < 0.05). RF, GM, and GA showed significant lower muscle activation in water environment regardless movement phases and steps per min (*p*< 0.01, *p* < 0.01 and *p* < 0.05, respectively). RF showed the greatest reduction of 44.8%MVIC in the ascending phase at 95 bpm while GM showed a maximal decrease of 51.7% MVIC in the ascending phase at 35 bpm. The decrease in muscle activation of BF is dependent on movement phases such that a significant lower muscle activation water is only observed in the ascending phase at all investigated steps per min (*p* < 0.05).

### 3.3. Comparison of Phases during FSU Exercise

Table 3 compares muscle activation at different movement phases in water and on land, respectively. The muscle activation of all tested muscles was higher in the ascending phase regardless of the exercise environment. However, such difference is not statistically significant in all tested muscles when considered a smaller effect size in water environment. Considering the movement phases and environments, among the target muscles, only BF and GM showed significant difference in muscle activation between movement phases on land (*p* < 0.05), while RF and GA showed no significant difference among movement phases in both environments. BF and GM also showed no significant difference among movement phases in water. On land, BF showed significantly higher muscle activation in ascending phase at all investigated steps per min (*p* ≤ 0.01), with the greatest increase of 31.4% MVIC at 35 bpm. GM only showed a significant difference at 35 bpm, with an increase of 35.8% of MVIC. In terms of %MVIC of all target muscles in different movement phases, although RF did not show significant difference in muscle activation among movement phases regardless the exercise environment, it can be seen that the highest %MVIC among the tested muscles, with 27.09% MVIC and 26.68% MVIC in the ascending and descending phase, respectively, in water.

### 3.4. Interaction between Each Muscle Performed Using Three Main Factors (Environment, Cadence, and Phase)

Regarding the various muscle activities examined, the analysis showed that the main effects for environments (*p* < 0.001; ES = 0.51), cadence (*p* < 0.001; ES = 0.657) and phases (*p* < 0.001; ES = 0.315) was significant in GM. Similarly, in RF the main effects for environments (*p* < 0.001; ES = 0.374), cadence (*p* < 0.001; ES = 0.738), and phases (*p* < 0.001; ES= 0.241) was significant. However, in BF, main significant effects could only be found in environments (*p* < 0.001; ES = 0.307) and cadence (*p* < 0.001; ES = 0.659). Among GA, a significant effect was found in environmental factor (*p* < 0.05; ES = 0.258), cadence (*p* < 0.001, ES = 0.831) and phases (*p* < 0.001; ES = 0.648). The environment* cadence* phases interaction was not significant in RF (*p* = 0.685; ES = 0.02), GA (*p* = 0.590; ES = 0.027) and GM (*p* = 0.229; ES = 0.075) while the interaction was approaching significant in BF (*p* = 0.055; ES = 0.142).

## 4. Discussion

This study hypothesized that lower limb muscle activation in water is lower than on land, and they increase along steps per min during FSU exercise. The results of this study provide evidence to support both hypotheses. All targeted muscles showed significantly lower levels of activation during the in water movements (*p* < 0.05), with RF showed a maximal reduction of 44.8%MVIC. Moreover, all target muscle, except GA, showed a significant increase in muscle activation as steps per min increased regardless of the environment (*p* < 0.05). The RF showed a more significant increase at all steps per min comparison (*p* < 0.01), with a maximum increase of 44.4% and 68.9% MVIC along the steps per min on land and in water, respectively.

### 4.1. The Effect of Steps per min on Lower Limb Muscle Activation

Our results showed that all tested muscles indicate an increasing trend of muscle activation, with a maximal increase of 44.4%, regardless of movement phases and environments. This study found that the %MVIC of three of the selected muscles (RF, GM, BF) at 95 bpm was significantly higher than that of 35 bpm, and the %MVIC of RF at 95 bpm was significantly higher than that of itself at all other steps per min comparisons, regardless of movement phases and environments. This result partly echoes with the finding of Zimmermann et al. [16], who investigated the effect of steps per min of stair-stepping exercise on lower limb muscle activation on land and found that the muscle activation of GM, RF, and GA increased significantly with steps per min. However, our results did not show similar trend in GA. Such difference is probably due to the difference of actions in studies. Our study required participants to stabilize their foot on the stool all the time, which GA acts as ankle stabilizer; while the research of Zimmermann et al. [16] involved action of stair climbing, GA acted as the prime mover during ankle plantar flexion. Our result also agrees with the finding of Lee and Lee [21], who found the muscle activation of RF increased as squatting speed increased on land. The similarity of results may be due to the comparable role of RF in both protocols. Lee and Lee [21] proposed that the increase in muscle activation along speed may be due to larger energy output in a shorter time. Clamann [22] found that increase in motor unit recruitment would increase the force output of a muscle, which aligns with the explanation of Lee and Lee [21].

Similar trend of change of muscle activation was also observed in the water environment and this trend may be attributed to the change of drag force. Drag force is the resistance of fluid acts on limbs movement and it increases as the speed of limbs movement in the same direction increases [2]. According to the drag force equation, increasing the speed by twice will increase four times of drag force [7]. Our result echoes with the finding of Chien et al. [6] that the RF muscle activation increased as steps per min increased from 30 bpm to 90 bpm. However, Chien et al. [6] found that the RF muscle activation at 90 bpm in water was comparable with that on land, while our study was unable to reproduce similar result. The discrepancy might be due to the difference of exercise design and angle of limb movement. The study of Chien et al. [6] involved the open chain knee extension movement of participants in a sitting position, which the lower limb movement was mostly contributed by the quadriceps muscle; while our study investigated the muscle activation in close chain FSU exercise, which participants might plantarflexed the ankle and thus reduced the muscle activation and demand on RF. In addition, our study standardized the height of stool instead of the starting position of knee angle at 90 degrees, which is the optimal angle to activate RF and thus, it may affect the muscle activation [23]. Moreover, our results are in line with Miyoshi et al. [24] that the muscle activation of hip extensor increased as the walking speed in water increases. Their study proposed that as hip extensor serves a propulsive function to push the body forward in water, greater water resistance will demand greater muscle activation of the hip extensor [24]. However, it is essential to consider that walking in water is a horizontal displacement while FSU is primarily a vertical displacement, the combination of different role of muscles and direction of water resistance may alter the pattern of muscle activation.

### 4.2. The Effect of Exercise Environment on Lower Limb Muscle Activation

Our result showed that performing FSU in water will elicit a lower muscle activation of GM, BF, RF and GA than that on land (*p* < 0.05), with a maximal decrease of 44.8% MVIC in the RF at 95 bpm ascending phase with a smaller effect size. This result echoes with finding of Yuen et al. [18], who investigated the muscle activation during squatting. The similar result may be attributed to exercise protocol—both squatting and FSU exercises are close chain. The reduction in muscle activation in water may be explained by the effect of buoyancy. Buoyancy, which equals to weight of the fluid displaced by body, is an upward force exerted by water which causes a reduction in body weight when immersed in water [25]. Participants of this study are immersed to the chest level in water and have 60% of weight reduction when performing FSU exercise in water [2]. Another possible cause could be neuromuscular system deactivation [26]. It is found that immersing participants to the chest level will induce weightlessness to muscle spindles, which decrease the activation of reflex of the pressure receptors within the spindles [26]. Thus, this proprioceptive effect impacts the neuromuscular system in terms of a muscle activation drop. The gravity changes in water may also impact multiple systems besides muscles, such as the vestibular and the visual system, by reducing the stimulation of gravireceptors [26,27]. However, there are uncertainties, such as how and to what extent hydrostatic pressure may affect muscle activation, and whether the kinematics are altered by the reduced ground reaction in water have yet to be discussed in papers [28].

### 4.3. Comparison of Ascending and Descending Phases of FSU

Our results showed that muscle activation of BF in all the steps per min and GM in 35 bpm were significantly higher during the ascending phase of FSU, with a maximal increase of 25.6% and 35.8%, respectively, while other muscles did not show significantly difference between the movement phases. On land, the range of %MVIC of BF activation found in this study is consistent with that of Ayotte et al. [12] who found that BF activation was about 12% MVIC during FSU. BF acted significantly more actively during the ascending phase than the descending phase on land. One of the possible reasons for higher activation of BF in ascending phase is that BF is a biarticular muscle responsible for hip extension and knee flexion [29]. During the ascending phase, BF acts as a hip extensor which leads to an increase in muscle activation while the descending movement may be facilitated by gravity which may lead to a decrease in muscle activation. In the ascending phase of FSU, the hip is flexed, and participants in our study were instructed to follow the steps per min according to auditory cues, so BF may contract eccentrically to control knee extension. These instructions may also pose greater activity to BF during the ascending phase only. Simenz et al. [13] also found that concentric activations of GM and RF are greater than that of eccentric in FSU. This finding is similar to ours to some extent by which only GM had significant difference between the two phases in 35 bpm. This finding can be explained by the difference in experimental set-up as the step height in their studies was higher (45.72 cm) and their participants were required to perform a loaded step-up by using six RM loads. More importantly, steps per min may be a determining factor of the significant difference as steps per min of FSU exercise was not instructed in their set-up.

In water, no significant difference was found between ascending and descending phases of the four muscles at any steps per min. Compared with the results on land, there is a discrepancy in the results in GM in 35 bpm and BF in all steps per min. The reason for the insignificant results remained unclear but could be a result of the interactions of the water properties with muscle activation. In our study, participants started the FSU exercise with immersion at the chest level and buoyancy can reduce the weight bearing by about 60% resulting in less muscle activation in the lower limbs [2,30]. The lowered muscle activation in BF was consistent with the findings from Yuen et al. [18] who revealed that BF had about 10% MVIC in both ascending and descending phases during water squatting. Additionally, buoyancy can generate an upward force to assist the upward motion leadings to a decrease in muscle activation of the hip extensors, i.e., BF and GM, during the ascending phase. Drag force may also hinder the knee extension movement because drag force will increase as velocity increases, which may replace BF in controlling the knee extension steps per min [7]. Thus, the muscle activation of GM in 35 bpm and BF in all steps per min in the ascending phase may not be significantly different from that of the descending phase.

The current results revealed insignificant interaction between environment, cadence, and phases among all the muscles despite the interaction was approaching significant in BF. Such insignificant findings might be due to the characteristics of the FSU exercise [31]. In the present study, the FSU exercise was performed with ascending and descending to maintain the exercise cadence with buoyancy supported promotes vertical displacement and resisted to promote resistance trainings. Further research investigating the interaction effects of FSU could be conducted under different factors.

### 4.4. Clinical Implications

Aquatic FSU exercise may be applicable to patients with lower limb weakness and in the initial phase of musculoskeletal rehabilitation. Our result showed that all tested muscle activation were lower in water, for example, the RF showed a maximal decrease of 44.8% MVIC in water when compared on land. This indicates that exercising in water is less demanding and suitable for patients with lower limb weakness. In addition, the physical properties of water, such as buoyancy and hydrostatic pressure, can reduce pain and swelling for patients in the early phase of post-operative period [32]. Moreover, aquatic FSU exercise can serve as a functional training for stair climbing, especially for patients with knee osteoarthritis, whose stair climbing ability are limited by quadriceps weakness and knee pain [33].

### 4.5. Limitations of Study

First, this study evaluated healthy population only, which limited the external validity as well as application to patients with musculoskeletal disorders. Second, participants were not restrained from engaging in other lower limb resistance training between land and water trials. It is possible that these influenced the results. Third, MVIC tests were conducted by manual methods. In other words, the maximal force resisted by participants to hold the testing positions is limited by the force of the operator. Nonetheless, in this study, all MVIC tests were conducted by the same instructor in order to minimize deviation.

## 5. Conclusions

This is the first study investigating the effect of steps per min on lower limb muscle activation during aquatic FSU exercise. This study has shown that GM, RF, BF, and GA muscle activation of healthy individuals increased with steps per min of FSU exercise, which RF showed a maximal increase of 44.4% MVIC. The exercise pattern of FSU exercise activates RF the most as RF acts a prime mover of knee extension and elevating the whole body. In addition, although the %MVIC of BF does not show significant difference in all steps per minute comparison, it serves as the prime mover of hip extension and controlling the speed of knee extension. Nevertheless, these muscle activations were lower in water than on land. The reduction in %MVIC in water is probably due to the buoyancy property of water, the increase in %MVIC along steps per minute is due to the drag force in water. However, the effect of hydrostatic pressure of water is not investigated in this study. Further investigations of factors, such as water immersion depth and angle of limb movement, are needed as these factors may affect the muscle activation pattern. Overall, aquatic FSU exercise can be a functional training exercise to improve stair-climbing ability. Moreover, the muscle activation of all tested muscles is much lower in water, indicating that exercising in water is less demanding and is suitable for patients with lower limb weakness. In future studies, researchers may recruit patients with knee osteoarthritis or knee pain to examine the effect of steps per min on their muscle activation.

## Figures and Tables

**Figure 1 healthcare-11-00441-f001:**
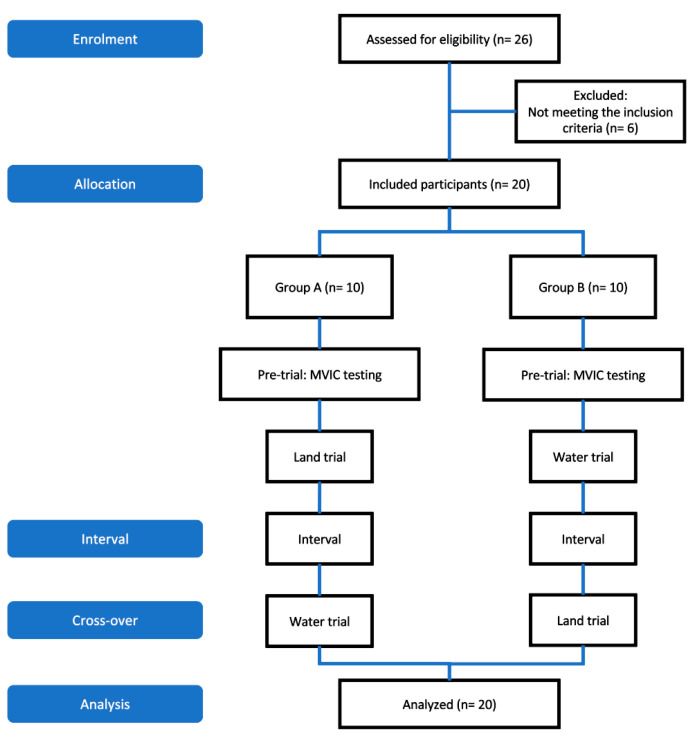
Study flow chart.

**Figure 2 healthcare-11-00441-f002:**
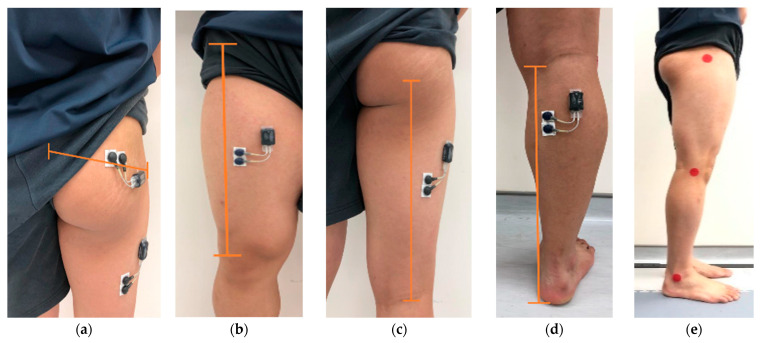
Location of the (**a**) gluteus maximus (GM) electrode placement, (**b**) rectus femoris (RF) electrode placement, (**c**) biceps femoris (BF) electrode placement, (**d**) gastrocnemius (GA) electrode placement, (**e**) three bony landmarks with markers attached.

**Figure 3 healthcare-11-00441-f003:**
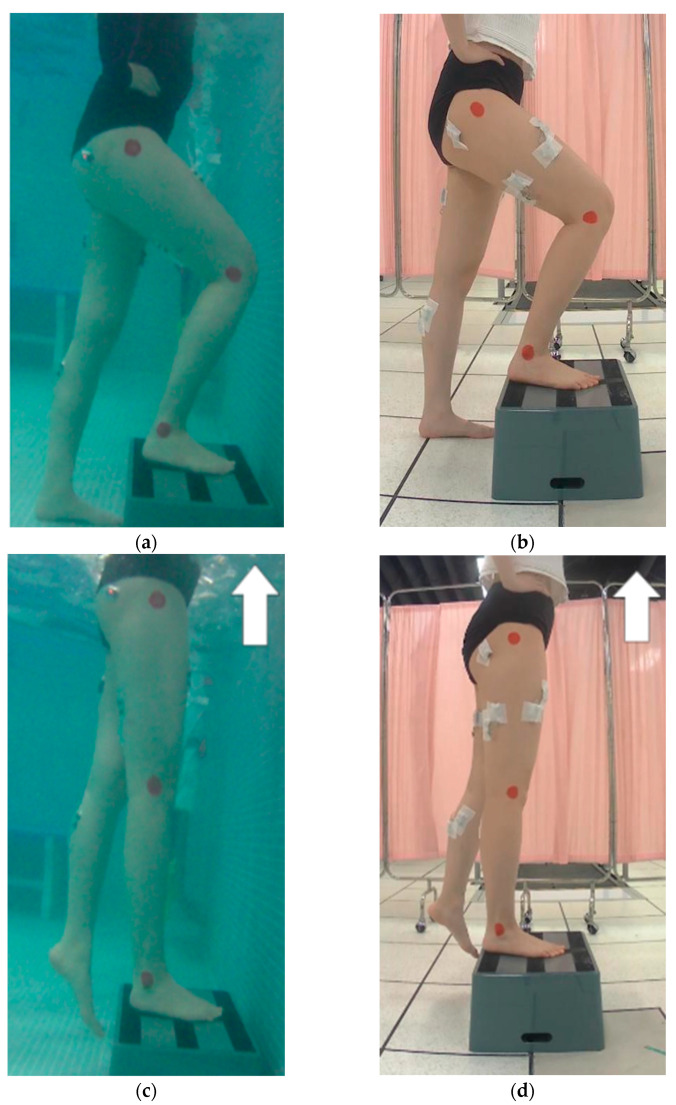
FSU exercise: starting position: (**a**) in water and (**b**) on land; ascending phase: (**c**) in water and (**d**) on land; descending phase: (**e**) in water and (**f**) on land.

**Figure 4 healthcare-11-00441-f004:**
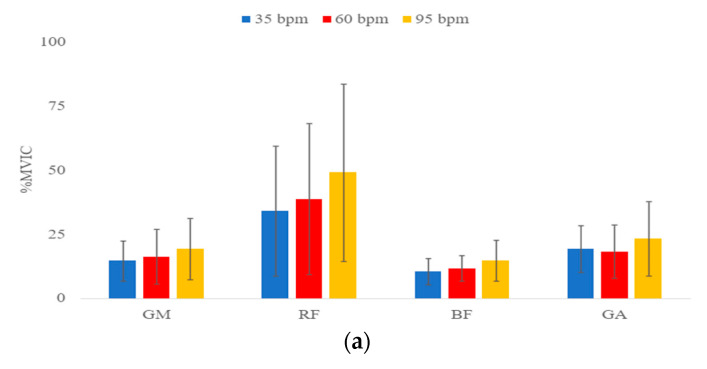
Means of percentage of maximum voluntary isometric contraction (%MVIC) for gluteus maximus (GM), rectus femoris (RF), biceps femoris (BF), and gastrocnemius (GA) performed during (**a**) ascending phase on land, (**b**) ascending phase in water, (**c**) descending phase on land, and (**d**) descending phase in water.

**Table 1 healthcare-11-00441-t001:** Descriptive characteristics of the 20 participants.

	n = 20
Gender (Male: Female)	10:10
Age (y)	21.1 ± 1.9
Weight (kg)	61.0 ± 7.7
Height (cm)	169.2 ± 6.0
BMI (kg/m^2^)	21.3 ± 1.8
Leg dominance (Left: Right)	0:20

**Table 2 healthcare-11-00441-t002:** The comparison of tested muscle activation at different media (GM = gluteus maximus, RF = rectus femoris, BF = biceps femoris, GA = gastrocnemius, A = ascending phase, D = descending phase).

Muscle	Steps per min (bpm)	Phase	On land (%MVIC, Mean ± SD)	In water (%MVIC, Mean ± SD)	*p* Value	95% Confidence Interval of the Difference		Effect Size
						Lower	Upper	
GM	35	A	14.6 ± 7.9	7.0 ± 4.8	0.000 *	−10.2	−3.9	−0.6
		D	10.7 ± 6.0	7.3 ± 3.7	0.009 *	−5.1	−1.0	−0.4
	60	A	16.2 ± 10.6	8.8 ± 4.5	0.003 *	−12.7	−2.5	−0.5
		D	13.4 ± 7.0	8.8 ± 4.1	0.001 *	2.3	7.0	0.9
	95	A	19.3 ± 11.9	11.9 ± 6.5	0.002 *	−10.8	−3.0	−0.5
		D	16.7 ± 10.2	10.3 ± 5.6	0.009 *	−10.7	−1.9	−0.4
BF	35	A	10.4 ± 5.1	7.6 ± 3.7	0.026 *	−5.0	−0.1	−0.6
		D	7.9 ± 3.9	7.1 ± 2.6	0.363	−2.7	1.0	−0.6
	60	A	11.7 ± 5.0	9.1 ± 3.9	0.011 *	−4.6	−0.5	−0.6
		D	9.3 ± 4.8	8.7 ± 4.3	0.697	−3.1	2.3	−0.6
	95	A	14.6 ± 8.0	10.7 ± 3.6	0.043 *	−7.6	−0.1	−0.5
		D	11.6 ± 6.1	10.2 ± 4.5	0.322	−4.4	1.6	−0.5
RF	35	A	34.0 ± 25.3	16.0 ± 9.7	0.000 *	0.4	5.1	0.5
		D	31.6 ± 21.5	15.9 ± 9.2	0.000 *	−1.0	2.6	0.2
	60	A	38.7 ± 29.4	19.6 ± 12.6	0.000 *	0.7	4.6	0.6
		D	37.0 ± 22.4	19.1 ± 11.8	0.000 *	−2.4	3.5	0.1
	95	A	49.1 ± 34.7	27.1 ± 18.3	0.001 *	0.1	7.7	0.5
		D	43.0 ± 19.7	26.7 ± 21.4	0.002 *	−1.5	4.4	0.2
GA	35	A	19.2 ± 9.2	14.7 ± 9.5	0.067 *	−10.5	0.7	−0.3
		D	23.7 ± 11.1	13.3 ± 7.4	0.003 *	4.1	16.7	0.8
	60	A	18.2 ± 10.5	12.7 ± 7.3	0.028 *	−10.7	−0.4	−0.4
		D	24.6 ± 14.1	12.5 ± 6.3	0.003 *	−18.2	−4.8	−0.5
	95	A	23.3 ± 14.0	16.0 ± 8.2	0.009 *	−11.7	−1.6	−0.4
		D	25.1 ± 10.3	15.9 ± 6.7	0.002 *	4.0	14.5	0.8

* indicates statistically significant difference (*p* < 0.05).

**Table 3 healthcare-11-00441-t003:** The comparison of tested muscle activation at different phases (GM = gluteus maximus, RF = rectus femoris, BF = biceps femoris, GA = gastrocnemius).

	Muscle	Steps per min (bpm)	Ascending Phase (%MVIC, Mean ± SD)	Descending Phase (%MVIC, Mean ± SD)	*p* Value	95% Confidence Interval of the Difference		Effect size
						Lower	Upper	
On land	GM	35	14.6 ± 7.8	10.7 ± 6.0	0.001 *	−5.5	−2.0	−0.8
		60	16.2 ± 10.6	13.4 ± 7.0	0.079	−7.3	0.8	−0.4
		95	19.3 ± 11.9	16.7 ± 10.2	0.191	−7.1	2.3	−0.3
	RF	35	34.0 ± 25.3	31.6 ± 21.5	0.218	−6.8	0.9	−0.3
		60	38.7 ± 29.4	37.0 ± 22.4	0.351	−7.8	4.4	−0.2
		95	49.1 ± 34.7	43.0 ± 19.7	0.167	−14.3	3.0	−0.3
	BF	35	10.4 ± 5.1	7.9 ± 3.9	0.000 *	1.6	3.4	1.4
		60	11.7 ± 5.0	9.3 ± 4.8	0.003 *	1.0	3.9	0.8
		95	14.6 ± 8.0	11.6 ± 6.1	0.013 *******	0.7	5.2	0.6
	GA	35	19.2 ± 9.2	23.7 ± 11.1	0.103	−9.9	1.0	−0.4
		60	18.2 ± 10.5	24.6 ± 14.1	0.167	−2.1	12.6	−0.3
		95	23.3 ± 14.5	25.1 ± 10.3	0.100	−0.9	6.9	−0.4
In water	GM	35	7.0 ± 4.8	7.3 ± 3.7	0.627	−1.7	2.2	−0.1
		60	8.8 ± 4.5	8.8 ± 4.1	0.794	−2.3	2.5	−0.1
		95	11.9 ± 6.5	10.3 ± 5.6	0.391	−3.7	1.1	−0.2
	RF	35	16.0 ± 9.7	15.9 ± 9.2	0.902	−2.1	2.4	0.0
		60	19.6 ± 12.6	19.1 ± 11.8	0.881	−2.0	2.2	0.0
		95	27.1 ± 18.3	26.7 ± 21.4	0.601	−5.9	3.3	−0.2
	BF	35	7.6 ± 3.7	7.1 ± 2.6	0.423	−0.8	1.9	0.2
		60	9.1 ± 3.9	8.7 ± 4.3	0.669	−1.3	2.0	0.1
		95	10.7 ± 3.6	10.2 ± 4.5	0.487	−1.0	2.0	0.2
	GA	35	15.0 ± 10.0	13.5 ± 7.5	0.502	−0.6	9.7	−0.0
		60	12.8 ± 7.3	12.6 ± 6.4	0.881	−2.6	2.0	−0.0
		95	16.1 ± 8.2	16.0 ± 6.6	0.931	−4.2	2.5	−0.1

* indicates statistically significant difference (*p* < 0.05).

## Data Availability

Not applicable.

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
