# Peer review of "Lower Limb Muscles’ Activation during Ascending and Descending a Single Step-Up Movement: Comparison between In water and On land Exercise at Different Step Cadences in Young Injury-Free Adults"

_healthcare, 2023, doi:10.3390/healthcare11030441_

Round 1

Reviewer 1 Report

Major concerns:

-        It would have been useful to measure the ratings of perceived exertion at the end of each exercise to provide a greater appreciation of the “physical intensity” of the different cadences. Also, the RPE scores/values have provides some comparison between the step-up and step= down in the water – to understand better the “physical effort” between these 2 types of movements. Further, this reviewer feels that a correlation can also be then conducted between levels of perceived physical effort and levels of muscle activity activated – and in this really will allow some real-world practical usefulness in determining real-time muscle activation.

-        It would been better to use the terminology of  “steps per min” rather than the current use of  “cadences”. Because this may “confuse” the reader for the current cadences of 35, 50 and 95 bpm do not actual equal to 35, 50 or 90 steps per min!

-        When filming in the water – dis the authors consider the refraction of the water when analysing the limbs movement. Also does the drag of the water affects the speed/velocity of the movements. Did the authors take these  2 conditions into account when filming in water.

-        Line 134. How was the values of MVC obtained from the mentioned movement or exercise tests, what equipment is used to assess the values of the isometric strength of the different muscle groups? What is the reliability of these isometric measures relative to the tester who conducted the tests? Authors need to explain in greater detail these tests/measures.

Minor issues:

-        Title. Can be more specific. Something like: “Lower limb muscles’ activation during ascending and descending a single step-up movement: Comparison between in-water and on-land exercise at different step cadences in young injury-free adults

-        Figure 1. Should the last 2 boxes of “Analysed (n = 10)” be joined together because you are analysing them altogether and not separately.

-        Line 157. Add the word “between” after the word “given”.

-        Figure 4 is unclear. Firstly, the each figure is too small to provide some clear resolution to their actual values. Also, the ascending on-land and ascending in-water should be place side-by-side of each other side to allow ease of comparison between these 2 movements values. Similarly, the descending on-land and descending in-water should be also side-by-side of each other.

-        Line 205 to 222. The results is explained in one single paragraph making it difficult for reader to follow closely what are the authors referring to. Suggest to break-this paragraph up into smaller bits by comparing between each “cadence” as one paragraph and so on to allow ease for reader.

-        For figures and tables. Use the term “on-land” and “in-water”. Also for percentage values, one decimal place will suffice.

-        Line 266. Change to: “All targeted muscles showed significantly lower levels of activation during the in-water movements …… ".

-        Throughout the manuscript, I think I sounds better to use the term “muscle activation” rather than “muscle activities”.

-        Line 269. Why the emphasis on RF muscles only because all other targeted muscles (except GA) also showed significantly greater activation levels with increased cadence.

-        Line 304. “trick movement”. I don’t think this is an appropriate term to use to explain this phenomena. Please re-write.

-        Line 347. Authors provided an example of sprinting here. However the movement in the present study was stepping-up. Please change and provide a better example.

Author Response

Dear reviewer,

Thank you for offering your precious time to give us very constructive feedbacks. Hope we can address your feedbacks properly. Happy New Year and Best Wishes.

Reviewer 2 Report

OVERVIEW:

The present study aimed to investigate the effect of progressive cadence on the surface electromyography (sEMG) of gluteus maximus (GM), biceps femoris (BF), rectus femoris (RF), and gastrocnemius (GA), when performing the Forward step-up exercise with different cadences in water and on land. The study is original and has interesting practical applications. The experimental design is sufficiently detailed and seems to be suitably executed. The results section needs to be improved regarding statistical analysis and data presentation. Therefore, several points should be revised and improved throughout the manuscript, as listed in the Specific Comments below.

SPECIFIC COMMENTS:

1.     Abstract: 

-  Line 19. The first use of the abbreviation %MIVC. Please, define it.

- Results. Please, revise the results according to the suggestions listed in the specific part. 

- Lines 23-25. The conclusion needs to be more clear. Please, rewrite it.

2.     Introduction:

- Line 35. Replace the references [2] and [3] with original studies that had measured the apparent weight in water instead of studies of revision.

Please, see and cite:

a) Harrison RA, Hillman M, Bulstrode S. Loading of the lower limb when walking partially immersed: implications for clinical practice. Physiotherapy, 1992; 78(3):164-166. doi: 10.1016/S0031-9406(10)61377-6

b) Alberton CL, Zaffari P, Pinto SS, Reichert T, Bagatini NC, Kanitz AC, Almada BP, Kruel LFM. Water-based exercises in postmenopausal women: Vertical ground reaction force and oxygen uptake responses. Eur J Sport Sci. 2021;21(3):331-340. doi: 10.1080/17461391.2020.1746835. 

- Line 48. Include the study by Alberton et al. (2011) when describing studies investigating lower limbs’ EMG between land and water-based exercises: 

Alberton CL, Cadore EL, Pinto SS, Tartaruga MP, da Silva EM, Kruel LF. Cardiorespiratory, neuromuscular and kinematic responses to stationary running performed in water and on dry land. Eur J Appl Physiol. 2011 Jun;111(6):1157-66. doi: 10.1007/s00421-010-1747-5. Epub 2010 Dec 3. PMID: 21127897. 

- Line 50. The detailed study to exemplify the environment effect should involve lower limbs’ EMG instead of upper limbs due to their characteristics involved in the body support in water vs. land. Please, substitute the Kelly et al. [7] with one that analyzed similar muscle groups/ patterns of movement to your study. 

3.     Materials and Methods

-  2.1. Study design. Line 76. The content presented in the flow chart shown in Figure 1 should be introduced in the study design.

-   2.1. Study design. Line 78. How were the participants allocated to group A or group B? Please, detail.

-   2.1. Study design. Figure 1. Several terms presented in this Figure were not explained in the text. For example, the information regarding the 26 participants assessed for eligibility needs to be added to the text. In addition, I suggest the change of the wash-out terminology to “interval”, which is more suitable for the study design. 

-  2.2. Sample size planning. Line 87. Include the numeric data of the effect size adopted for the sample size calculation.

- 2.3. Participants. Line 92. Please, detail how and where the participants were recruited.

- 2.3. Participants. Line 100. Please, remove Figure A1 corresponding to the IPAQ. Use only the corresponding reference in the text.

- 2.5.1. Skin preparation, electrode placement and joint markers. Line 119. Please, rewrite the sentence: “According to previous guidelines [14,16]...” by “According to previous studies [14,16]…” because the cited references are not “guidelines”.

- 2.7. Data processing. Line 189. Why were kinematic data collected? I suppose that filming was performed with the specific purpose of synchronizing the EMG data. However, this explanation needs to be clearer. Which procedure was adopted for defining the initial and final time for each selected step? How was the filming synchronized to the sEMG signal??? Why was raw kinematics not presented?

- 2.8. Statistical Analysis. Line 194. The authors must entirely revise the statistical analysis. I suggest transforming the non-parametric data by natural log for using only parametric analysis. The ANOVA three-way should be performed for each muscle, using the three main factors (environment, cadence, and phase) and their interactions. Alternatively, the GEE is a semi-parametric test and could be applied.

4.     Results

- The same results are repeatedly present in different tables. One table should present the means and SD (or 95%CI), considering all main factors. The pairwise comparison needs to be expressed in the table by symbols. Exact p-values for each factor and interaction need to be presented (e.g., do not use < 0.05, except for p<0.001).

- Which type of effect size was presented? How was it calculated? Which are the reference values for their interpretation?

- Specific comments about the results will be presented in the next round after the authors revise the statistical analysis.

5.     Discussion

- Specific comments about the discussion of the results will be presented in the next round after the authors revise the statistical analysis.

- Since effect size is presented for each main factor comparison, the authors should also discuss them. For example, for environment comparison, I strongly suggest presenting a % reduction in the EMG for each muscle (or a % range for different cadences). This information is helpful for the practical applications of the results.

- 4.1. The effect of cadence on lower limb muscle activities. Please, include the following studies that used similar patterns of movements in the cadences comparison for water-based exercises:

a) Alberton CL, Pinto SS, Cadore EL, Tartaruga MP, Kanitz AC, Antunes AH, Finatto P, Kruel LF. Oxygen uptake, muscle activity and ground reaction force during water aerobic exercises. Int J Sports Med. 2014 Dec;35(14):1161-9. doi: 10.1055/s-0034-1383597. 

b) Pinto SS, Cadore EL, Alberton CL, Silva EM, Kanitz AC, Tartaruga MP, Kruel LF. Cardiorespiratory and neuromuscular responses during water aerobics exercise performed with and without equipment. Int J Sports Med. 2011 Dec;32(12):916-23. doi: 10.1055/s-0031-1283176. 

- 4.2. The effect of exercise environment on lower limb muscle activities. The specific BF results (i.e., a significant difference between environments only during the ascending phase but not in the descending one) need more attention and explanation. In lines 323-325, the correct % of weight reduction needs to be used (as in the introduction comment, please, do not use studies of review), considering specificities of population and depth immersion.

6.     Conclusion

-       The conclusion should involve all analyzed factors and all muscle specificities.

Author Response

Dear reviewer,

Thank you so much for offering your precious time to give us very useful and constructive feedbacks. Hope we can address your comments properly and happy to take any advice on board. Happy new year and best wishes. Cheers!

Round 2

Reviewer 1 Report

Minor issues:

- Table1 should be placed under section 2.3 and not in the results Section. 

- Figure 4. Too small. please increase in size for better resolution. Also, all columns are missing the SD value (T-sign). 

Author Response

Dear reviewer,

Thank you for your feedback once again. Please see attached.

Best wishes. 

Reviewer 2 Report

Overview:

The manuscript is significantly improved. However, after the first review, I still have some critical concerns, especially related to the analysis and presentation of the results.

Specific Comments

  1. Line 56. You should describe a study investigating lower limbs' EMG to replace the study by Kelly et al. In the current version, the authors exemplified ground reaction force at different speeds [12], which was not the study's goal.
  2. Lines 108-110. The authors included the reference for IPAQ questionnaire, which is ok. Nevertheless, I still reinforce that the entire IPAQ should be wholly removed from the article (Appendix A).
  3. 2.7. Data processing. The explanation regarding the kinematic data processing should be included in the manuscript.
  4. 2.8 Statistical Analysis. I am still in doubt if the authors performed a suitable statistical analysis because there are several errors in the description of the tests and results presentation. For example, why the student's t-test was used if you used three-way ANOVA for repeated measures? In the results section, authors still describe: "Results of Friedmann test…". A specific Interactions section is presented, but their real meaning is not explained. Why interactions between two (of the three) factors is not presented? The significant interactions are the main analysis for these statistics and based on their results, we need (or not) to do separate post hoc analyses per each factor. Therefore, I strongly recommend a consultation with a statistician to perform the statistical analysis and help with the presentation of the results.
  5. The response related to the effect size should also be included in the manuscript. 
  6. Table 2. Where is the GA muscle? I still believe table 2 could be suppressed. Symbols should present the pairwise between cadences in Figure 4. The effect size may be presented along the text with the interpretation of the results.
  7. Line 376. Please, read the article [12] and include the correct percentage of reduction observed in the study.

Author Response

Dear reviewer,

Thank you for your time. See if this revision meets your requests. Thank you very much.

Best Wishes. 

Round 3

Reviewer 2 Report

I ams satisfied with the changes in the current version.

Author Response

Dear reviewer,

Thank you very much for your acceptance for this revision. Best Wishes!